# Chromatin Dynamics and Transcriptional Control of Circadian Rhythms in *Arabidopsis*

**DOI:** 10.3390/genes11101170

**Published:** 2020-10-06

**Authors:** Aida Maric, Paloma Mas

**Affiliations:** 1Centre for Research in Agricultural Genomics (CRAG), CSIC-IRTA-UAB-UB, Campus UAB, Bellaterra, 08193 Barcelona, Spain; aida.maric@cragenomica.es; 2Consejo Superior de Investigaciones Científicas (CSIC), 08028 Barcelona, Spain

**Keywords:** Arabidopsis, circadian clock, chromatin, transcriptional rhythms

## Abstract

Circadian rhythms pervade nearly all aspects of plant growth, physiology, and development. Generation of the rhythms relies on an endogenous timing system or circadian clock that generates 24-h oscillations in multiple rhythmic outputs. At its bases, the plant circadian function relies on dynamic interactive networks of clock components that regulate each other to generate rhythms at specific phases during the day and night. From the initial discovery more than 13 years ago of a parallelism between the oscillations in chromatin status and the transcriptional rhythms of an *Arabidopsis* clock gene, a number of studies have later expanded considerably our view on the circadian epigenome and transcriptome landscapes. Here, we describe the most recent identification of chromatin-related factors that are able to directly interact with *Arabidopsis* clock proteins to shape the transcriptional waveforms of circadian gene expression and clock outputs. We discuss how changes in chromatin marks associate with transcript initiation, elongation, and the rhythms of nascent RNAs, and speculate on future interesting research directions in the field.

## 1. The Plant Circadian Clock

The circadian clock is an endogenous timing mechanism able to generate biological rhythms with a period of 24 h. Although the circadian system is daily synchronized by changes in light and temperature, it is also capable of sustaining the circadian oscillations under constant environmental conditions [1]. The circadian clock regulates an ample range of physiological, developmental and metabolic processes ensuring that they are appropriately phased in concordance with the cellular demands [2]. The mechanisms responsible for the generation of the rhythms are quite complex and involve the orchestrated expression and function of key essential components [3,4]. This molecular oscillator is exquisitely connected with synchronizing cues to timely drive the rhythms of the biological processes or rhythmic outputs controlled by the clock [5].

The circadian function is particularly important in plants, possibly due to their sessile nature and the need for constantly monitoring the environment for proper adaptation and survival. Indeed, the anticipatory function of the plant circadian system has been proposed to provide an adaptive advantage and improve fitness [6,7,8,9]. At its basis, a common mechanism responsible for the generation of rhythms in eukaryotic cells relies on circadian negative feedback loops of activators that drive the expression of negative components, which feedback to inhibit their own expression [10]. This basic transcriptional regulatory mechanism is complemented by additional layers of circadian control including among others chromatin regulation, RNA metabolism and changes in cellular and subcellular localization [11,12,13,14].

The molecular components of the plant circadian system have been extensively identified and characterized in the model plant *Arabidopsis thaliana*. Circadian studies on crops have also started to uncover the divergences but also the similarities of clock components in plant model systems and other crops [15,16]. The firstly identified *Arabidopsis* clock component, *TIMING OF CAB EXPRESSION (TOC1)* or *PSEUDO RESPONSE REGULATOR1 (PRR1)*, was initially isolated nearly 25 years ago [17], and characterized as an evening-expressed gene, important in the regulation of circadian rhythms and flowering time [18,19,20,21,22,23]. Two additional clock components, CIRCADIAN CLOCK ASSOCIATED 1 (CCA1) and LATE ELONGATED HYPOCOTYL (LHY), were later identified and characterized [24,25]. CCA1 and LHY are two morning-expressed and partially redundant single MYB-containing proteins that form heterodimers to fulfill their circadian function [26,27]. TOC1, CCA1, and LHY were proposed to form a transcriptional feedback loop essential for circadian rhythmicity in *Arabidopsis* [20].

Research over the last years has contributed to extend our knowledge on the molecular circadian network, adding numerous new components and regulatory mechanisms [28]. For instance, other members of the *PSEUDO RESPONSE REGULATOR* (*PRR*) gene family including *PRR3*, *PRR5*, *PRR7*, and *PRR9* [29,30] were found to be closely associated with the *Arabidopsis* clock [31,32]. The *PRR* gene family members are sequentially expressed, starting with *PRR9* with a peak-phase closed to dawn, followed by *PRR7* and *PRR5* at midday and by *TOC1* with a peak-phase of expression at dusk [19,29]. Other evening-expressed components include the *EARLY FLOWERING 3* (*ELF3*) and *ELF4* genes coding for two plant-specific proteins without recognizable domains [33,34] and *LUX ARRHYTMO* (*LUX*) coding for a single MYB-like GARP transcription factor [35,36]. These three clock proteins were first identified by genetic screens for components involved in flowering and hypocotyl regulation [33,37,38,39]. The proteins were found to interact forming a multi-protein complex called the EVENING COMPLEX (EC) [40,41]. Recent studies have indicated that the EC components may also have independent functions from the EC [42,43,44].

The main clock components engage in highly complex regulatory networks that ensure the specific phase of oscillator gene expression during the day and night (Figure 1). For instance, CCA1 and LHY repress the *PRR*s [45], which in turn suppress *CCA1* and *LHY* transcription [31]. TOC1 not only represses *CCA1* and *LHY* expression [46,47,48] but nearly all the oscillator genes by binding to their promoters [48]. The *EC* components are repressed by CCA1 and LHY in the morning [49,50,51] and by TOC1 in the evening [48]. In turn, the EC acts as a transcriptional repressor directly binding to the *PRR9*, *PRR7*, and *LUX* promoters and repressing their expression [52,53,54,55]. By repressing the repressors of *CCA1* and *LHY*, the EC indirectly promotes *CCA1* and *LHY* expression.

The identification of the regulatory function of the clock components uncovered a prevalent number of repressors, opening the question about the mechanisms of circadian transcriptional activation. Over the past recent years, a number of new components have been proposed to function as activators of clock gene expression (Figure 1). One example includes the *LIGHT-REGULATED WD1* (*LWD1*) and *LWD2* genes, encoding WD (Trp and Asp)-containing proteins [56]. The LWDs directly bind to the promoters of *CCA1*, *PRR9*, *PRR5*, and *TOC1* to activate their expression [57,58]. Another example includes REVEILLE 8 (RVE8 also known as LHY-CCA1-LIKE5 or LCL5), a protein that belongs to the CCA1 and LHY single-MYB protein family [59,60]. Despite being members of the same family of plant transcription factors, RVE8 activates the expression of *TOC1* and *PRR5*, thus, in an opposite way to the repressing function of CCA1 and LHY [59,61]. RVE8 also directly activates the expression of *PRR9*, *ELF4* and *LUX* [62]. Other members of the RVE protein family appear to be functionally redundant with RVE8, as the *rve4rve6rve8* triple mutant accentuates the long period phenotype of *rve8* single mutant [62].

Overall, it is well established that generation of 24-h rhythms requires the accurate coordination of the expression and activities of numerous clock components. These components regulate each other through multiple regulatory mechanisms to ultimately control plant physiology and development in synchronization with the environment [12]. In this review, we focus on one of these regulatory mechanisms: chromatin changes and its connection with circadian transcriptional regulation. We do not attempt to provide an exhaustive description of all what is known related to the topic but rather to provide an update on the most recent and relevant discoveries functionally linking chromatin status and the plant circadian clock. Readers are encouraged to consult recent reviews that have in-depth descriptions on this and other specific topics [14,63,64].

## 2. Transcriptional Dynamics and Chromatin Status

Transcriptional regulation is intimately connected with chromatin status, which can be modified by changes in DNA methylation [65], histone covalent modifications [66,67], nucleosome remodeling and replacement of core histone with histone variants [68] and higher-order chromatin location and organization [69]. The four core histones (H2A, H2B, H3, and H4) can be modified at different amino acid residues by a repertoire of modifications such as acetylation, methylation, phosphorylation, ubiquitination, sumoylation, glycosylation, ADP ribosylation, biotinylation, and carbonylation [67]. These modifications are able to alter the accessibility of chromatin to the transcription machinery, thus influencing the transcriptional outcome [66].

Acetylation of histones is controlled by histone acetyltransferases (HATs) and has been mostly associated with gene activation [70]. Plants have multiple HATs [71], which have been functionally characterized to a different degree. One major class of plant HATs present homology with the yeast and Tetrahymena GCN5 (GENERAL CONTROL NONDEREPRESSIBLE 5) family [72]. GCN5 appears to be important in the regulation of many processes including cell differentiation, organogenesis, and responses to light and cold [73]. The acetylation of histones can be reverted by histone deacetylases such as RPD3 (REDUCED POTASSIUM DEPENDENCY PROTEIN 3)-like and SIR2 (SILENT INFORMATION REGULATOR PROTEIN 2)-like, which are conserved across all eukaryotes [70,71].

Histones can be also methylated by histone methyltransferases (HMTs) including a group of SET (SU(VAR)/E(Z)/TRX) domain proteins. Histone methylation is associated either with gene activation or with repression depending on the amino acid residue of the modification [74]. For instance, histone H3 methylation of lysine 4 (H3K4) or lysine 36 (H3K36) is generally associated with activation of gene expression, whereas methylation of lysine 9 (H3K9) or lysine 27 (H3K27) is usually related to heterochromatin and gene repression [74]. Histone methylation is also reversible through the action of histone demethylases such as lysine-specific demethylase 1 (LSD1) and Jumonji C (JmjC) domain-containing proteins, which play important roles in the regulation of plant growth and development [75].

## 3. Functional Link between Chromatin and Circadian Transcription

The connection between chromatin changes and the *Arabidopsis* circadian clock was first reported about thirteen years ago [76]. The study showed that the rhythmic changes in *TOC1* mRNA expression were associated with parallel oscillations in histone acetylation [76]. The trough of *TOC1* expression coincided with histone deacetylation and with maximal CCA1 repressor binding [76]. Later studies reported that other histone marks also associate with the chromatin state at the *TOC1* promoter [77,78]. The accumulation of some of these histone marks such as histone 3 lysine 4 trimethylation (H3K4me3) was found to antagonize the binding of clock repressors, ensuring that repression occurred at the proper time during the day and night cycle [78]. The rhythms in histone marks were found not only at the *TOC1* promoter but in many oscillator loci [63]. From that point on, a number of chromatin-related factors were identified as “writers” and “erasers” of the histone marks important for the circadian oscillation. Here we describe the most relevant studies over the last couple of years (Table 1).

It is well established that increasing patterns of histone acetylation at the promoters of clock genes correlate with their rising phase of expression [63]. A recent study has provided some clues about chromatin-related factors contributing to this histone acetylation [79]. The study has shown that the expression of *HAF2*, a histone acetyltransferase of the TAFII250 family 2, is activated at midday, and this activation promotes histone acetylation at the *PRR5* and *LUX* loci, coincident with their raising phase of expression [79]. If histone acetylation associates with activation, what are the chromatin-related components that facilitate the histone deacetylation during the declining phase? A number of histone deacetylases had been already identified [71]. However, a recent report has provided evidence that the evolutionarily conserved Sin3-histone deacetylase complex (HDAC) is connected with the plant clock [80]. The study showed that components of the Sin3-HDAC complex, SAP30 FUNCTION-RELATED 1 (AFR1), and AFR2, are circadianly-regulated. Moreover, the evening-expressed AFR proteins contribute to the repression of *CCA1* and *PRR9* during the night, facilitating histone deacetylation by directly binding to their promoters. Thus, rhythmic histone deacetylation by the Sin3-HDAC complex contributes to shape the appropriate circadian waveforms of morning-expressed circadian genes [80].

Other histone deacetylases have been identified in studies of evening-expressed genes such as *TOC1,* which is also regulated by changing histone deacetylation patterns [81]. Indeed, HISTONE DEACETYLASE 9 (HDA9) and ELF3 directly interact and regulate the declining phase of *TOC1* after dusk. This regulation relies on the direct binding of HDA9 to the *TOC1* promoter through the interaction with ELF3. The EC-HDA9 complex facilitates histone deacetylation and represses *TOC1* expression during the night. The components of the EC also interact with HDA9 and with HIGH EXPRESSION OF OSMOTICALLY RESPONSIVE GENE 15 (HOS15), a WD40 repeat protein at the promoter of the clock- and flowering-related gene *GIGANTEA* (*GI*), leading to histone deacetylation and transcriptional repression of *GI* [82]. PRR9 also interacts with a member of the plant Groucho/Tup1 corepressor family, TOPLESS/TOPLESS-RELATED (TPL/TPR) and with HDA6, defining together a repressive complex at the promoters of *CCA1* and *LHY* [90]. The studies thus provide examples of the direct interaction between clock components and chromatin-related factors, which underscores the importance of chromatin status and regulation of circadian clock gene expression.

In addition to histone acetylation/deacetylation, other histone marks are also associated with circadian gene expression. For instance, the circadian accumulation of H3K4me3 at clock loci was proposed to be mediated by the histone methyltransferase SDG2/ATXR3 (SET DOMAIN GROUP 2/ARABIDOPSIS TRITHORAX RELATED 3) [78]. In a more recent study, Song et al. have not only verified the role of SDG2/ATXR3 controlling circadian histone methylation but also identified a role for the Jumonji C domain–containing histone demethylase (JMJ14) as regulator of circadian oscillations [85]. Notably, the study has reported a feedback between histone modifications and the diurnal regulation of circadian clock genes [85]. On one hand, the histone methyltransferase SDG2 (as a “writer”) and the histone demethylase JMJ14 (as an “eraser”) regulate the expression of circadian oscillator genes by modulating H3K4me3 accumulation. In turn, CCA1 and LHY were shown to regulate directly the diurnal expression of *JMJ14* and indirectly that of *SDG2*, which leads to the rhythmic patterns of H3K4me3 accumulation in the target loci. Furthermore, a genome-wide analysis showed a limited overlap between H3K4me3 and H3K9ac marks in morning-phased and evening-phased genes, suggesting specific roles of different histone modifications controlling diurnal gene expression in *Arabidopsis*. Another study has recently shown that the expression of *TOC1* is also repressed by histone demethylation [83]. In this case, the repression requires the coordinated interaction of CCA1/LHY with the Lysine-Specific Demethylase 1 (LSD1)-like histone demethylases, LDL1 and LDL2 [75]. LDL1 and LDL2 also interact with the histone deacetylase HDA6 providing a double mechanism for repression of *TOC1* expression by both histone demethylation and deacetylation [83]. Notably, the authors have also recently shown that HDA6 and LDL1/2 can in turn interact with TOC1, and the complex contributes to the repression of *CCA1*, *LHY*, and other circadian related genes [84]. Therefore, the LDL1/2-HDA6 complex seems to play a relevant role controlling the expression of a subset of clock-related genes.

ELF3 also represses target gene expression at the end of the day by directly interacting with a protein from the chromatin-related SWI2/SNF2-RELATED (SWR1) complex [89]. The SWR1 complex associates with chromatin and catalyzes the histone variant H2A.Z exchange at genomic sites. H2A.Z is a well-conserved histone variant [91] that influences transcriptional activities of associated genes [92]. Consistently, the EC-SWR1 complex is able to bind to the *PRR7* and *PRR9* loci to control both the deposition of H2A.Z and the repression of these genes at dusk. The study thus provides a mechanism by which repressive chromatin domains are temporally defined by the circadian clock [89].

A majority of studies on the transcriptional circadian regulation has mainly focused on steady-state mRNA expression. A recent report however, has provided evidence on the rhythms in transcriptional synthesis, circadian nascent RNAs and chromatin modifications [86]. The study showed a modular function of RVE8, with its MYB domain responsible for the DNA binding, and its LCL domain providing the platform for the interaction with the clock components known as NIGHT LIGHT-INDUCIBLE AND CLOCK-REGULATED proteins (LNKs) [61,93,94,95]. LNKs rhythmically recruit the RNA Polymerase II and the transcript elongation FACT complex to co-occupy the promoters of the clock genes *TOC1* and *PRR5* [86]. The RVE8-LNKs interaction and the recruitment of the transcriptional machinery ultimately define not only transcript initiation and elongation but also the chromatin status including changes in histone marks such as H3K4me3 accumulation. Analyses of nascent RNAs by nuclear run-on transcription by bromouridine immunocapture indeed showed that the rhythmic occupancy of the transcriptional machinery results in oscillatory nascent RNAs [86].

Another recent study has provided further information on transcript elongation and pre-mRNA processing of *CCA1*. The study focused on histone H2B monoubiquitination (H2Bub), which in *Arabidopsis* is controlled by HISTONE MONOUBIQUITINATION1 (HUB1) and HUB2 E3 ubiquitin ligases together with the UBIQUITIN-CONJUGATING ENZYME 1 (UBC1) and UBC2 E2-conjugating enzymes [88]. HUB proteins interact with the previously uncharacterized RNA-binding motif-containing proteins, SPEN3 and KHD1 [88]. In the *spen3-1* and *hub1-4* mutants, H2Bub accumulation was reduced and *CCA1α* and *CCA1β* splice isoforms were altered. The mutant plants showed short circadian period length phenotypes in agreement with the reduced expression of *CCA1*. Overall, the study showed that H2Bub deposition associated with *CCA1* transcript elongation and pre-mRNA processing are two processes that are facilitated by the HUB1/HUB2 complex [88].

The circadian clock controls many outputs or rhythmic biological processes that occur at the most appropriate diurnal or seasonal time. Chromatin marks have been recently associated with seasonal regulation [96]. Indeed, genome-wide analyses in a natural population of perennial *Arabidopsis halleri* have uncovered a close connection of histone H3 lysine 27 trimethylation (H3K27me3) deposition with the control of seasonal gene regulation. The seasonal accumulation of H3K27me3 is phase-delayed in comparison with the H3K4me3 oscillation, most prevalently for genes associated with environmental memory. The authors thus proposed that H3K27me3 marks can control seasonal responses by monitoring past transcriptional activity for long-term regulation of expression in a subset of genes in plants grown under natural environmental conditions [96].

One fundamental clock output is the photoperiodic regulation of flowering time [97,98]. Many studies have previously shown the importance of chromatin remodeling at flowering-related loci [99]. A recent study has proposed a hierarchical graphical model inferring genome-wide gene regulatory networks connecting flower development and circadian signaling [100]. The study identified two major connecting hubs: HFR1 (LONG HYPOCOTYL IN FAR-RED) and LHY. Indeed, the network analyses showed that LHY controls a number of transcription factors directly related with flower development [100]. Notably, during the transition to flowering, *LHY* shows in turn a significant change in H3K4me3 at the shoot apical meristem [101]. HFR1 directly interacts with the histone acetyltransferase HAC1 (HISTONE ACETYLTRANSFERASE 1) and bind to *AG* (AGAMOUS, a floral development factor) to activate its expression via histone acetylation. Consistently, the authors found a flower-specific peak of H3K27ac at the *AG* gene body closely coinciding with a HFR1 binding motif. HFR1 plays a key role in the transducing signals from light and temperature to influence circadian signaling and flowering development. It would be interesting to apply this kind of approaches with time series to further infer dynamics and new connections between chromatin changes at the core of the oscillator and in clock related outputs.

Another example connecting chromatin changes with the regulation of flowering time was recently provided by a study on the florigen gene *FLOWERING LOCUS T (FT)*. *FT* shows a 24-h oscillation under long-day (LD) conditions with a peak of expression during the day. At dusk, the HISTONE DEACETYLASE 2C (HD2C) is recruited to the *FT* locus and deacetylates histones to repress *FT* transcription. HD2C competes with CONSTANS (CO), the activator of *FT*, for the binding of the *MORF-RELATED GENE 2* (*MRG2*) [102]. Thus, the study involves a histone deacetylase and histone methylation readers to shape the photoperiodic-dependent waveform of *FT* expression. H2B monoubiquitination and SPEN3 function are not only important for *CCA1* transcript elongation and pre-mRNA processing as mentioned above [88], but are also important in the regulation of the flowering [88]. Indeed, the *spen3-1* mutant plants showed a delay in flowering time that correlated with an enhanced expression of the flowering-related gene *FLOWERING LOCUS C* (*FLC*), most likely due to an increased distal versus proximal ratio of its antisense *COOLAIR* transcript [88].

*FT* is regulated by the precisely coordinated action of several players. For instance, CO forms a protein complex together with the B and C subunits of Nuclear Factor Y (NF-Y) to activate *FT* expression close to dusk. In contrast, the Polycomb repressive complex 1 (PRC1) and PRC2 proteins silence *FT* expression. PRC proteins show H3K27 methyltransferase activity that generates H3K27 trimethylation (H3K27me3) and maintain this mark, also facilitating other repressive marks [103]. A recent study has shown that the NF-CO complex favors a reconfiguration of the chromosomal conformation at *FT* resulting in reduced binding of Polycomb proteins to the *FT* promoter [104]. This chromatin looping and reduced binding of Polycomb proteins relieve the Polycomb-mediated silencing, resulting in *FT* de-repression near dusk.

Another example includes the role of histone demethylation in the regulation of flowering time. The study shows that JMJ13, which possesses H3K27me3 site-specific demethylase activity, acts as a flowering repressor, and modulates flowering time in a photoperiod- and temperature-dependent manner [87]. The study also shows that the expression of main clock genes such as *LHY* and *CCA1* and flowering-related genes such as *CO* was up-regulated in *jmj13* mutant plants [87]. These results open the question of whether JMJ13 directly regulates clock through changes in histone demethylation at their loci. JMJ5/JMJ30 has been also connected with circadian regulation and in particular with temperature compensation [105]. However, this function appears not to involve changes in H3K36 methylation at the circadian clock loci [105].

A recent study has shown that the circadian clock regulates other outputs such as seed dormancy through the concerted action of the ATP-dependent chromatin-remodeling factor PICKLE (PKL) [106] and the EC component LUX [107]. The two proteins interact and bind to the locus of the *DELAY OF GERMINATION1* (*DOG1*) gene, which encodes a protein involved in seed dormancy [108]. The H3K27me3 accumulation at the *DOG1* locus was reduced in *pkl* or *lux* mutants. The authors conclude that the circadian clock, through LUX and its interaction with PKL, modulates seed dormancy during seed development by controlling the expression of *DOG1*. This regulation might be important to prevent seeds from becoming overly dormant [107].

## 4. Future Perspectives

Circadian studies are rapidly expanding our view on how the circadian system works in different parts of the plant, and how mobile signals are able to synchronize clocks in distal organs [109,110,111,112,113]. Over the past recent years, it has become increasingly clear that circadian information is shared through short- and long-distance communication. The strength of circadian cell-to-cell coupling differs among cells and tissues [114,115]. For example, coupling is minimum among cotyledon cells [116], variable in leaves [117,118,119], high in roots [120] and between the vasculature and neighbor mesophyll cells [121], and very high within cells at the shoot apex [122]. Long-distance circadian synchronization on the other hand, seems to occur through shoot-to-root photosynthetic signaling [123], light piping down the root [124] and by the movement of ELF4 from shoots to regulate the period of the root clock in a temperature-dependent manner [113]. The studies highlight specific and autonomous circadian function, which urgently calls for studies on changes of the chromatin status not only with a temporal resolution (circadian or seasonal) but also with spatial definition in order to identify cell, tissue-, and organ-specific circadian chromatin landscapes.

Likewise, over recent years, chromatin conformation capture approaches have provided an unprecedented three-dimensional view of chromatin organization [125]. Studies with animal cells have uncovered a hierarchical system with compartment, domains and loops, playing important roles in the control of transcription [126]. Similar studies in plants have now shown that plant cells contain comparable high-order structures [69] with the notable exception of the TAD-like loop domains found in mammals or the lack of a plant CTCF-like insulator protein [126]. It would be then interesting to fully understand the functional divergences of the high-order chromatin formation and organization in plants compared to animals. Responses of plant chromatin conformation to different environmental and cellular signals would be also interesting to elucidate, focusing on the functional connection between their formation and their specific biological functions. Circadian changes on chromatin conformation and nuclear localization in different tissues and organs would be also worth exploring. We surely have ahead many interesting discoveries within the plant circadian field.

## Figures and Tables

**Figure 1 genes-11-01170-f001:**
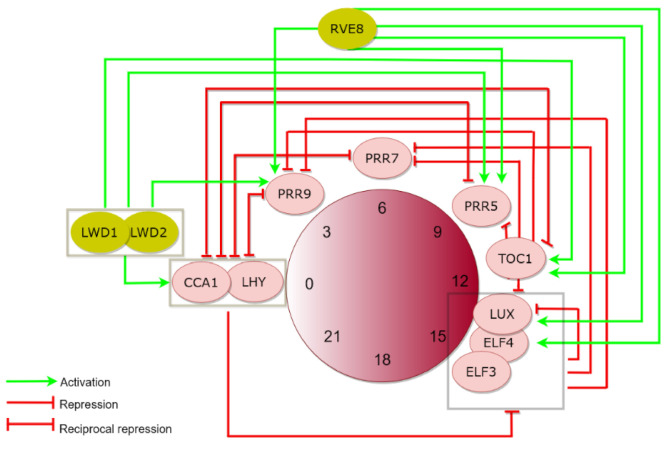
Schematic drawing depicting the basic regulatory network at the core of the *Arabidopsis* circadian oscillator. Oscillator genes are expressed at different phases during the day and night, from morning hours on the left on the central circle (clock) to evening hours on the right. Repression is depicted as red lines ending in small perpendicular dashes whereas activation is indicated by the green arrows. Clock components that interact to perform their regulatory function are encircled in grey line boxes. CCA1 and LHY are repressors of *PRR* genes, including *TOC1.* PRRs in turn repress the expression of *CCA1* and *LHY.* Expression of *Evening Complex (EC)* components (*LUX*, *ELF4*, *ELF3*) is repressed by both CCA1/LHY and TOC1. The EC represses expression of *PRR9*, *PRR7*, and *LUX.* The regulatory network is dominated by repressive interactions, although recent studies have uncovered a number of activating factors such as LWD1/2 and RVE8, which activate the expression of multiple morning- and evening-expressed oscillator genes. Please consult the main text for further details.

**Table 1 genes-11-01170-t001:** List of the most recent findings connecting chromatin changes and circadian oscillator genes in *Arabidopsis.*

Histone Mark	Clock/Chromatin-Related Factor	Regulated Clock Component	Reference
Acetylation	HAF2	*PRR5, LUX*	[79]
Deacetylation	Sin3-HDAC	*CCA1, PRR9*	[80]
	ELF3-HDA9	*TOC1*	[81]
	EC-HDA9-HOS15	*GI*	[82]
	HDA6-CCA1/LHY	*TOC1*	[83]
	HDA6-TOC1	*CCA1, LHY*, other clock genes	[84]
Methylation	SDG2 (ATXR3)	*CCA1, LHY*	[78,85]
	RVE8/LNKs	*PRR5, TOC1 nascent RNAs*	[86]
Demethylation	JMJ14	*CCA1, LHY*	[85]
	CCA1/LHY-LDL1/2	*TOC1*	[83]
	TOC1-LDL1/2	*CCA1, LHY, other clock genes*	[84]
	JMJ13	*CCA1, LHY*	[87]
Monoubiquitination	HUB1/HUB2	*CCA1*	[88]
Histone Variant H2A.Z	ELF3-SWR1	*PRR7, PRR9*	[89]

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
