# Peer review of "Chromatin Dynamics and Transcriptional Control of Circadian Rhythms in Arabidopsis"

_genes, 2020, doi:10.3390/genes11101170_

Round 1
Reviewer 1 Report
In this review, the authors update the latest studies on the molecular mechanisms underlying chromatin structural changes that are thought to be deeply correlated with robust circadian oscillations. It covers broad range of important findings almost thoroughly, mainly focusing on histone modifications. Since such a concise review have been waited, it will actually be welcome by experts in many fields. I think the review is recommended for publication essentially as it is.
minor points
1) P4L112: rv4 -> rve4
2) P4L145: SU(VAR)/E(Z)TRX -> SU(VAR)/E(Z)/TRX
3) P7L267: Here, the authors take up two factors, HFR1 and LHY. Function of LHY in chromatin remodeling at flowering loci should be described in this paragraph.
Author Response
We appreciate the Reviewer #1 comments and thank him/her for the revision of our review. We have modified it following the reviewer indications. More specifically:
1) We have corrected rve4.
2) We have corrected SU(VAR)/E(Z)/TRX.
3) We have briefly explained the link of LHY-chromatin-flowering although there are not many studies, to our knowledge, addressing this particular topic.
Reviewer 2 Report
This manuscript is a very clearly written review of the chromatin dynamics and transcriptional control of the Arabidopsis circadian rhythm. The authors have thoroughly reviewed the current literature and clearly identifies opportunities for new research directions.
I have a recommendation to add a reference showing that PRR9, PRR7, and PRR5 represses CCA1 and LHY genes through recruiting transcriptional corepressors TOPLESS proteins and HDACs (doi.org/10.1073/pnas.1215010110). This report will give a critical link between chromatin dynamics and the clock transcription factors.
Author Response
We appreciate the Reviewer #2 comments and thank him/her for the revision of our review. We have modified it following the reviewer indications. More specifically:
1) We have included and briefly explained the study showing the connection of PRRs with TPL and HDA6.